# Raman Spectroscopy for the Time since Deposition Estimation of a Menstrual Bloodstain

**DOI:** 10.3390/s24113262

**Published:** 2024-05-21

**Authors:** Alexis Weber, Anna Wójtowicz, Renata Wietecha-Posłuszny, Igor K. Lednev

**Affiliations:** 1Department of Chemistry, University at Albany, State University of New York, 1400 Washington Avenue, Albany, NY 12222, USA; aweber@albany.edu; 2Laboratory for Forensic Chemistry, Department of Analytical Chemistry, Faculty of Chemistry, Jagiellonian University, 2 Gronostajowa St., 30-387 Kraków, Poland; anna.wojtowicz@doctoral.uj.edu.pl (A.W.); wietecha@chemia.uj.edu.pl (R.W.-P.)

**Keywords:** Raman spectroscopy, menstrual blood, time since deposition, chemometrics, bloodstain aging

## Abstract

Forensic chemistry plays a crucial role in aiding law enforcement investigations by applying analytical techniques for the analysis of evidence. While bloodstains are frequently encountered at crime scenes, distinguishing between peripheral and menstrual bloodstains presents a challenge. This is due to their similar appearance post-drying. Raman spectroscopy has emerged as a promising technique capable of discriminating between the two types of bloodstains, offering invaluable probative information. Moreover, estimating the time since deposition (TSD) of bloodstains aids in crime scene reconstruction and prioritizing what evidence to collect. Despite extensive research focusing on TSD estimations, primarily in peripheral bloodstains, a crucial gap exists in determining the TSD of menstrual bloodstains. This study demonstrates how Raman spectroscopy effectively analyzes biological samples like menstrual blood, showing similar aging patterns to those of peripheral blood and provides proof-of-concept models for determining the TSD of menstrual blood. While this work shows promising results for creating a universal model for bloodstain age determination, further testing with more donors needs to be conducted before the implementation of this method into forensic practice.

## 1. Introduction

Forensic chemistry is a rapidly developing field of analytical chemistry. Blood is one of the most frequently found traces discovered at crime scenes [1]. Most often, bloodstains originate due to the loss of peripheral blood caused by tissue damage from an assault. However, in some types of crimes, such as sexual assault, menstrual bloodstains can also be present at the scene.

During crime scene reconstruction, the biological origin of the stain, peripheral versus menstrual blood, may indicate a completely different context of the event in question. While the presence of peripheral blood may be indicative of an assault, menstrual blood can be present due to innocuous reasons [1,2]. Therefore, it is necessary to have a technique that can identify and analyze both peripheral and menstrual blood with statistical confidence. Due to their similar appearance, it is difficult to visually distinguish between these two types of bloodstains after the bloodstain has dried [3]. So, it is essential to have a methodology that can analyze all bloodstains at a biochemical level. Previous research by our lab has shown Raman spectroscopy is a technique capable of discriminating between peripheral and menstrual blood [3]; moreover, this technique provides more probative information. 

Additional information that can help investigators to reconstruct the event in question is obtained by determining the time since deposition of the evidence in question. For biological evidence, this is conducted by determining the time since deposition (TSD) of the stain. Beyond assisting investigators in understanding the timeline of the crime, TSD calculations can be used to streamline the evidence collection process. At the scene of a violent crime, the amount of potential evidence can be overwhelming due to the existence of unimportant biological fluids that were present before the crime occurred. The collection of all materials at a scene can lead to the collection of extraneous evidence, creating more samples for the forensic scientists to analyze. To alleviate this bottleneck, TSD determinations can be used to streamline the forensic workflow so that only the most pertinent evidence is collected and further analyzed.

At a biochemical level, both peripheral and menstrual blood differ in their composition. Peripheral blood consists predominantly of blood cells (red blood cells, white blood cells, and thrombocytes) and biomolecules, including hemoglobin, fibrinogen, albumin, glucose, immunoglobulins, and tryptophan [2]. Menstrual blood, in addition to possessing peripheral blood, is also composed of vaginal fluid and cells from the epithelia, basal lamina, uterine glands, and stroma. These cells in menstrual blood are the product of the shedding and expulsion of the functional endometrial layer of the uterus during menstruation [4,5]. Another noteworthy difference is the clotting properties of peripheral blood compared to menstrual blood. During an injury, peripheral blood will form clots to stop the bleeding to ensure that the victim does not bleed out. The process of clotting is caused by proteins and enzymes within the coagulation pathway. In contrast, menstrual blood does not clot and shows a greater fibrinolytic activity [6]. This is because, during menstruation, clotting would be harmful to the individual. Additionally, peripheral blood varies in composition from menstrual blood due to the presence of matrix metalloproteinases (MMPs). MMPs are proteolytic enzymes involved in menstrual blood used in the reconstruction and breakdown of tissues during the menstrual cycle [6]. Finally, the components urea, lactic acid, lactate, and acetic acid have also been reported in menstrual blood [3].

The difference in composition is the basis for the biochemical methods used during analysis to distinguish these two blood types. Methods include the detection of MMPs, estrogen receptors, and fibrinogen [7], as well as measuring the degradation products of fibrinolysis [8]. Differences in properties resulting from a different composition are, in turn, the basis of discriminant methods using vibrational spectroscopy, such as Raman spectroscopy [3] and attenuated total reflection Fourier transform infrared (ATR-FTIR) spectroscopy [9,10,11], which have been successfully used to distinguish between menstrual and peripheral bloodstains. However, the main task during a forensic investigation is not only to identify bloodstains, but also to extract from them as much probative information as possible about the event. By analyzing the location of the stains, known as blood pattern analysis (BPA), it is possible to determine how the bloodstains were created [12]. Additionally, a DNA analysis performed on the stains can be used for the final identification of the origin of the individuals involved [13]. However, the largest challenge during forensic work is to determine how much time elapsed between the occurrence of the crime and the discovery of the scene, which refers to the stains’ TSD [14]. The accurate estimation of the TSD enables the verification of witnesses’ testimonies, limits the number of suspects, and assesses alibis. Additionally, it can help investigators to identify which bloodstains are relevant to the crime and reconstruct the course of the event.

Many methods to determine the TSD of bloodstains have been tested, but there is still no universally agreed upon method that has been incorporated into forensic practice. The previously tested methods used encompass a wide range of destructive and non-destructive techniques and are summarized in several reviews [14,15,16,17]. Recently, as indicated within the cited reviews, there has been a shift from destructive methods based mainly on chromatographic procedures to non-destructive methods based on various types of spectroscopy methods, including ultraviolet–visible (UV-Vis) spectroscopy, fluorescence spectroscopy, reflectance spectroscopy, vibrational spectroscopy methods (infrared and Raman spectroscopy), atomic force microscopy, and electron spin resonance spectroscopy.

Research on the TSD of bloodstains has mainly focused on peripheral bloodstains. However, in previous work, we proposed a method to probe the aging of both peripheral and menstrual bloodstains by steady-state fluorescence spectroscopy [18]. Both biological fluids exhibited similar kinetic changes during aging, which were assigned to the presence of endogenous fluorophores: tryptophan, nicotinamide adenine dinucleotide (NADH), and flavins. Blood exposure to the oxygen-rich external environment caused the chemical phenomena of tryptophan oxidation, NADH accumulation, and initial flavin adenine dinucleotide (FAD) deficiency followed by flavin oxidation. These changes caused a decrease in tryptophan fluorescence intensity and an increase in the fluorescence strength of NADH and flavins [18,19].

Notwithstanding the success of fluorescence spectroscopy research, vibrational spectroscopy has demonstrated the broadest analytical range of any technique used to study bloodstain aging. Furthermore, the technique has numerous other advantages, such as sensitivity, rapidness, reproducibility, accuracy, portability, non-invasiveness, cost-effectiveness, and minimum sample preparation [17]. It is therefore not surprising that the methods of IR spectroscopy [20,21] and Raman spectroscopy [22,23,24] are strongly explored for TSD applications in recent studies. In a previous study, peripheral bloodstains were aged one week post-deposition, with continued research presenting an aging model for blood aged for two years [23]. Doty et al. created a regression model that was able to estimate the TSD of bloodstains with statistical confidence. However, at a crime scene, there is a possibility that menstrual blood can be present in tandem with peripheral blood. Therefore, it is imperative that forensic investigators be able to analyze menstrual blood with the same accuracy as peripheral blood. Raman spectroscopy has already been shown to be capable of discriminating between peripheral and menstrual bloodstains [2]. And once the source of the stain is known, the next step would be to determine how long the stain has been present. This will allow for the corroboration of witness statements with scientific evidence.

The use of vibrational spectroscopy techniques for determining the TSD of stains is primarily based on detecting changes in the main component of blood, hemoglobin (Hb), an oxygen-carrying protein that accounts for 97% of the dry blood content [14]. When blood exits the body, the whole volume of Hb saturates with environmental oxygen and transforms into oxyhemoglobin (oxyHb). OxyHb is then further autoxidized into methemoglobin (metHb). During this process, the Fe^2+^ molecules that are present in the heme groups are converted into Fe^3+^, which causes Hb to lose the ability to bind oxygen. In vivo, the activity of two enzymes, glutathione peroxidase and cytochrome b5 reductase, allows metHb to be converted back to oxyHb. However, under ex vivo conditions, these enzymes are no longer active and oxyHb autoxidation is irreversible [14,15,16,17,25]. Therefore, metHb is further transformed into hemichrome (Hc), the denatured form of Hb that results from the rearrangement of the Hb protein chains. The further transformation of Hc involves the aggregation of the degradation products that are formed during the aging process [14,15,16,17,26]. The gradual transformation of individual forms of hemoglobin after bloodstain deposition is accompanied by a change in properties that can be detected using spectroscopic methods. Therefore, the oxyHb → metHb → Hc conversion is commonly used as a method of determining TSD for bloodstains [20,21,22,23,24,27].

In recent years, Raman spectroscopy has been developed for bloodstain aging analysis [17]. The Lednev lab has worked on developing and commercializing a universal method for the identification of all main body fluids using Raman spectroscopy via the start-up company, SupreMEtric LLC, Rensselaer, NY, USA. Regarding the determination of the TSD of bloodstains, research has primarily focused on analyzing peripheral bloodstains [22,23,24]. At a crime scene, both peripheral and menstrual bloodstains can be present. As such, it is pertinent to understand if the capabilities of Raman spectroscopy can estimate the TSD of menstrual bloodstain as this method does with peripheral bloodstains. To fill this gap in the research, a preliminary study on a method for estimating the TSD of menstrual bloodstains, based on Raman spectroscopy, was created. Changes in the Raman spectra of menstrual bloodstains were monitored for over two weeks post-deposition under ambient conditions. And the menstrual bloodstains were measured at 12 time points: 1, 5, 9, 24, 48, 72, 96, 120, 144, 168, 236, and 336 h. Using Raman spectroscopy paired with chemometrics, we were able to estimate the time since deposition of menstrual blood. To the best of our knowledge, this proof-of-concept study is the first published work on the aging of menstrual bloodstains with the use of Raman spectroscopy.

## 2. Materials and Methods

For this study, two samples of fresh menstrual blood were collected from different Caucasian donors using a disposable menstrual cup on the third day of the donor’s menstrual cycle. The samples were collected following a protocol approved by the Institutional Review Board (IRB) at the University at Albany. Before the blood collection, all donors signed a written consent. This consent included an acknowledgement that the donors were healthy adults, not utilizing prescription or recreational drugs. Prior to the deposition of blood, aluminum-covered microscope slide substrates were prepared by cleaning them using alcohol, rinsed with deionized water, and then dried to remove any contaminants present. After collection, 20 µL of the samples was deposited onto the substrate using a micropipette and aged under ambient conditions. To measure changes to the sample post-deposition, the menstrual bloodstains were analyzed at 12 time points over a period of two weeks. The time points measured were 1, 5, 9, 24, 48, 72, 96, 120, 144, 168, 236, and 336 h.

### 2.1. Instrumental Collection Parameters

Raman spectra were collected using a Horiba XploRA™ Plus Confocal Raman Microscope (HORIBA Scientific, Piscataway, NJ, USA), with a 785 nm excitation laser at 10% power and a 50× long working distance objective. The instrument was calibrated using a silicon standard (peak at 520.6 cm^−1^) prior to the experimental collection each day. For each spectrum, the collection parameters included 15 accumulations with 10 s exposure times over the spectral range of 300–1800 cm^−1^. Utilizing an automatic mapping stage, a 12-point map was collected from the area of 36 µm × 32 µm, keeping the same step size between points. The area was chosen by using the optical image to find a part of a bloodstain that was uniform and flat. This allowed the laser beam to remain in focus on the bloodstain while also accounting for the bloodstain’s heterogeneity.

At each prearranged post-deposition time point, a Raman spectral map was prepared from 12 individual spots to account for the sample heterogeneity. Each spectral map was obtained from a fresh, previously unirradiated area on the bloodstain. As each map was collected from a small region, it was imperative to ensure that the quality of the spectra was consistent. For each map, the individual spectra were analyzed. In total, only two low-quality spectra were removed based on a significant deviation from an average spectrum, which was caused by sample heterogeneity. Bloodstains, specifically menstrual bloodstains, are heterogenous in nature, and some spots will possess more fluorescence than others. For this experiment, spectra that highly deviated from the average spectrum due to detector saturation because of a large background were removed.

### 2.2. Statistical Analysis

Data preprocessing and analysis was performed in MATLAB (MathWorks, Inc., Natick, MA, USA; version 9.3.0.713579, R2017b) equipped with PLS_Toolbox version 8.7 (Eigenvector Research Inc., Wenatchee, WA, USA). Prior to the statistical analysis, the outlier spectra were identified and confirmed by the use of principal component analysis (PCA) loadings and score plots. Prior to any TSD modeling, all Raman spectra were preprocessed. The data were truncated to 315–1800 cm^−1^; the baseline was corrected by use of the Automatic Whittaker Filter (AWF) algorithm (lambda = 850, *p* = 0.001) and normalized by the total area; and the mean was centered. A PCA was utilized for the preliminary evaluation of the TSD data for menstrual blood. Two PCA scores plots were generated in this study, one for each donor. Each of the PCA plots were created using 4 principal components (PCs), and each class within the scores plot was based on the analysis hour post-deposition.

For estimated TSD predictions, two classification models were created using the mean centered data. A partial least squares discriminate analysis (PLSDA) model was produced. This model was created using the same calibration and external validation data. Samples from donor 1 were used for calibration, utilizing Venetian blinds for internal cross-validation (CV), while samples from donor 2 were used for external validation. For model creation, the PLSDA model was created using 3 latent variables (LVs). The PLSDA classification models were generated using binary division for the determination of the relative freshness of a menstrual bloodstain, classifying the stains as fresh (<72 h) or older (>96 h). This was used as a form of preliminary analysis in estimating the TSD of a menstrual bloodstain.

After the broad estimation of the TSD of menstrual bloodstains, a more specific model was used to predict the age of a bloodstain to within an hour range. To build a predictive model for TSD extrapolation, a partial least squares regression (PLSR) analysis was utilized. The PLSR model was created using 5 LVs. For this analysis, donor 1 was used for the calibration data set and for internal cross-validation. The results from donor 2 were utilized for the external validation. When considering the results from this analysis, the ideal plot would have a minimal divide between the spectra at each time point. The analysis of menstrual bloodstains using both the classification and regression models can provide an accurate TSD determination for the samples analyzed using Raman spectroscopy.

## 3. Results and Discussion

Menstrual blood is a complex biological sample that is composed primarily of blood and vaginal fluid [5]. The composition of menstrual blood will vary within a donor based on many factors, including the day of the menstrual cycle, amount of bleeding during menstruation, concentration of hormones (including estrogen), and amount of vaginal fluid present [5,28]. Upon the deposition of the fluid, the differences in menstrual blood from peripheral blood were noticeable until the sample dried. The menstrual blood samples were less viscous than those of peripheral blood as well as lighter in color. However, during the aging process, the menstrual blood samples behaved visually similarly to those of peripheral blood, changing from a red fluid to a dark brown stain over time. In this proof-of-concept study to investigate the aging of menstrual bloodstains, samples were collected from two donors and prepared by depositing 20 μL of fluid onto aluminum foil-covered microscope slides. The stains were kept under ambient conditions and analyzed on the Horiba Xplora Plus Raman spectrometer (HORIBA Scientific, Piscataway, NJ, USA) at 1, 5, 9, 24, 48, 72, 96, 120, 144, 168, 236, and 336 h (two weeks) post-deposition. In this section, the changes in the spectra of menstrual blood over time and the use of chemometric models to estimate the time since deposition are discussed.

### 3.1. Menstrual Bloodstain Raman Spectra

When examining all the spectra collected in a single map, there were no apparent differences in the Raman bands, but they did exhibit varying fluorescent background intensities. Additionally, the spectra were consistent with the previous results by Virkler et al. [29], which established that a bloodstain is chemically heterogenous and shows slight spectral variations from one spot of the sample to another. The averaged Raman spectra of menstrual bloodstains obtained at the 12 time points over the first two weeks post-deposition, acquired from donor 1, are shown in Figure 1A.

As seen in Figure 1A, an overall increase in the fluorescent background was observed during the two weeks of menstrual blood aging. However, the changes in the fluorescent background, relative to the time since deposition, were not always monotonic and varied for the two donors, which could be because the bloodstain size and thickness were not controlled, and the quantitative contribution of fluorescence varies with the position of the focused laser beam on a solid heterogeneous sample. A similar characteristic increase in the fluorescent background was previously observed by Doty et al. [22] when peripheral bloodstains were aged for one week after deposition. This phenomenon was assigned to hemoglobin transformation as hemoglobin derivates, which are known to contribute to a high-fluorescence background [30], form after blood exposure to the external environment as the bloodstain ages. In addition, heme aggregation and the consequent changes in its electronic structure contribute to the fluorescence background changes with the time since deposition [31]. However, the changes in the fluorescence background observed for menstrual blood in this study were more significant than those reported for peripheral bloodstains. This indicates that there is a unique distinction in the aging process of menstrual and peripheral bloodstains, which can be attributed to the difference in their biochemical composition [5,19].

The averaged preprocessed Raman spectra are displayed in Figure 1B. These spectra are consistent with those previously reported for menstrual blood [2]. Peak assignments for individual Raman bands were determined based on an extensive literature search and are summarized in Table 1. As previously reported by Sikirzhytskaya et al. [2], the Raman spectrum of menstrual blood is very similar to that of peripheral blood, differing only in the relative intensity of some peaks. Peripheral blood possesses stronger band intensities for the peaks correlating to tryptophan (746 cm^−1^), hemoglobin’s heme group (1124 cm^−1^), and the amide III vibrational mode of the polypeptide backbone (1248 cm^−1^). The Raman band related to the amide I vibrational mode (1660 cm^−1^) possesses a characteristic higher relative intensity in the menstrual blood Raman spectra [2]. 

Though menstrual blood contains vaginal fluid, it has previously been shown that the Raman spectrum of vaginal fluid clearly differs from that of blood, indicating that the components of vaginal fluid contribute very little to the menstrual blood Raman spectrum [3]. The spectral assignments attributed to the bands of the vaginal fluid components are marked in green in Table 1. This tentative assignment includes vaginal fluid components such as urea, lactic acid, lactate, and acetic acid [33]. Thus, the peaks in the Raman spectra of menstrual bloodstains were primarily assigned to the vibrational frequencies originating from the blood components (marked in red in Table 1). It is well-documented that the near-IR Raman spectrum of blood is dominated by the contribution from heme hemoglobin [29,31]. Its additional components include other proteins, DNA bases, and other essential biomolecules [3,22,23,31,32,34,35].

### 3.2. Menstrual Bloodstain Raman Spectra

As blood is a complex matrix, it can be difficult to identify a singular chemical phenomenon responsible for changes in an individual Raman peak with the TSD. As it is evident from Figure 1B, there are several apparent changes to the spectra of menstrual blood overtime. In order to study the kinetics of the changes, the peaks’ intensity was plotted versus time. Mono-exponential decay functions, described by the equation: y = A exp(−x/t) + y0, were fitted. The most significant changes were observed for three bands at 890, 1369, and 1577 cm^−1^ and are presented in Figure 2. The kinetic data obtained for both donors were similar in terms of qualitative changes; however, some differences were observed in the quantitative comparison of the fitted function parameters, as can be seen in Table 2. As we have previously reported [18], it is not surprising that the characteristic decay time varies between the two donors, because it depends on the specific biochemical composition of the sample, which could vary significantly for menstrual blood samples both because of the donor and the day of the menstrual cycle.

Due to the dominant contribution of the main protein in blood, hemoglobin (Hb), changes in the intensity of Hb derivatives’ marker bands should be first indicated. The most prominent alternations were seen at 1369 cm^−1^, the band which is related to methemoglobin (metHb) vibrations. This metHb band showed an exponential increase in intensity with a time constant in the range of 41–67 h observed over the aging trial. This observation is supported by the fact that, after being deposited and aged ex vivo under ambient conditions, oxyhemoglobin (oxyHb, oxygen-bound hemoglobin form) is naturally autoxidized to metHb and cannot be re-converted due to the lack of free cytochrome-b5 reductase as it is an enzyme responsible for converting metHb back to oxyHb [14].

Significant changes were also observed in the band at 1577 cm^−1^, which can be assigned to heme but may also contain some contributions of protein and DNA base vibrations. For this band, an overall exponential increase in intensity with an average decay rate defined by a time constant of 81–113 h was observed throughout the experiment. The monotonic course of these changes suggests that this peak may also act as a useful indicator of the age of menstrual bloodstains in the range from 1 to 336 h post-deposition.

Regarding non-heme blood components, peak intensity changes in the band 890 cm^−1^ should be mentioned. During the first 24 h post-deposition, a clear exponential decrease in the intensity of this band with the rate of changes ranging from 25 to 47 h was noted. The band at 890 cm^−1^ was previously assigned to the vibrations of essential biomolecules, such as polysaccharides and amino acids, and additionally may be attributed to lactate from vaginal fluid [33]. Additionally, a relative intensity increase was apparent at the 1448 cm^−1^ band. This band is attributed to the tryptophan present in the blood [16]. Changes in this amino acid residue may also be associated with exposure to a high-oxygen-rich external environment, which can cause the denaturation and aggregation of hemoglobin [1]. In the spectra of menstrual blood, the shift of the Raman band from 752 to 745 cm^−1^ after 24 h and the increase in the intensity of the band at 1448 cm^−1^ in the first 9 h post-deposition were observed in Figure 1B. The changes in both these bands are characteristic of structural changes, such as denaturation and aggregation, in tryptophan-containing proteins [16]. However, further studies are needed to determine specifically how tryptophan changes overtime in bloodstains.

The visual analysis of the Raman spectra clearly indicates significant monotonic changes in the individual spectral bands with respect to the TSD. Although the sample heterogeneity reduces the accuracy of determining the TSD based solely on the individual Raman bands in the next step, a multivariate statistical data analysis was performed, and appropriate models were built to estimate the TSD of menstrual bloodstains.

### 3.3. Calculating the Time since Deposition

The score plot for donor 1 is presented in Figure 3.

After the spectral examination was completed, a principal component analysis (PCA) model was created to test the behavior of the data. PCA is an unsupervised statistical technique that is used to group data based on their similarities. Principal components (PCs) are generated based on the variance within the data set [36]. For each donor, the individual preprocessed spectral data was input and the PCA model built with the resulting scores plot produced using the four PCs, which accounted for the most variation between classes. Each point within the PCA score plot represents an individual spectrum, and the classes for the model were defined as the time point post-deposition. 

The PCA models were used to understand how the spectra were grouped compared to each other, and excellent preliminary results were obtained. The data demonstrated that grouping by the hour post-deposition, though divided between classes, was more distinctive at the earlier hours (1-, 5-, 9-, 24-, and 48-h post-deposition). After 72 h, the noticeable clustering was reduced, and the remaining spectral data points overlapped. This division around the 72-h mark occurred in the PCA scores plots for both donors. This indicates that, when using supervised classification models, it is difficult to create classes for the model based on the individual hours post-deposition. Therefore, the subsequent classification models were created using groups of multiple hours post-deposition to estimate the TSD of menstrual bloodstains.

After the PCA analysis, a partial least squares discriminant analysis (PLSDA) model was created. Since this is a supervised technique, this model was built using a calibration data set where the classes were specified. After this, new data, defined as the test data set, were input into the model, which then predicted how the spectral data were obtained [36,37]. Based on the PCA results, the PLSDA model was created using a binary division, ≤72 h and ≥96 h. The PLSDA model was created using three latent variables and the confusion matrix results are shown in Table 3. Venetian blind cross-validation (CV) was used for the internal testing of the model’s usability. For the external validation, spectral data from a new donor was used to test the validity of the model. The PLSDA model was created for binary discrimination when estimating the window for the TSD of menstrual bloodstains. During an investigation, this can act as a preliminary screening method for the rapid determination of whether a menstrual bloodstain is fresh (<72 h) or old (>96 h). This differentiation will allow the investigators to know which stains are relevant for analysis and allow them to proceed with the further examination of a smaller group of evidence. By confirming the origin of the bloodstain and knowing if a sample is fresh or old will indicate which samples need to undergo additional analyses. This thus allows only the most informative samples to move forward for DNA profiling. Hopefully, this will increase the overall efficiency of the forensic investigative process.

The PLSDA model performed well and had a 92% accuracy in estimating the TSD range. A further investigation of these results was conducted to understand how each hour, within the binary model, was predicted during classification. A 40% classification threshold was used to determine if there was a misclassification for each hour. Misclassified hours are indicated as a red text in Table 4 for the external validation results. For both the internal and external validation results, the 72-h spectra were the only ones that were misclassified. This is not a surprising result, as 72 h is on the boarder of the binary model division. Within this area of the results, it was expected that there would be some bleeding between the classes.

Even after the preliminary determination of whether a stain is fresh or old is completed, it is still important to establish a more precise TSD. This was conducted using a regression model. Partial least squares regression (PLSR) is a supervised statistical technique that finds linear changes within the spectra as a function of time [36,37]. As this is a supervised technique, both a calibration data set and an external test data set were required to create the model. The external validation results of the regression model are shown in Figure 4. The R^2^ values for the calibration, internal CV, and prediction (external validation) were 0.98, 0.92, and 0.90 respectively. This indicates that the regression model created has great promise for predicting the TSD of a menstrual bloodstain down to an hourly window. This thus shows that utilizing PLSR in tandem with the PLSDA can act as a method for determining the TSD of menstrual bloodstains, when analyzed using Raman spectroscopy.

The results of the PLSR model are presented in a logarithmic scale in Figure 4 that minimizes the error visual appearance. The average TSD predictions are also plotted on a linear scale, with the error bars showing one standard deviation, in Figure 5. These results indicate that the error range for predicting the TSD of menstrual blood increases after 72 h post-deposition. However, the relative error remained overall consistent during the aging trial. This is consistent with previous results of calculating the TSD of peripheral blood under ambient conditions [23]. This indicates that, as blood ages, the confidence window for calculating the TSD of blood diminishes due to the variable aging of bloodstains. This is even more evident within menstrual bloodstains, relative to peripheral blood, due to the varied composition of menstrual blood, which is dependent on the donor [5,18,19]. Based on the results of the TSD predictions, it was determined that the PLSR model is the most promising for determining the age of the bloodstains. This is because this method takes into consideration the entire spectral region, as compared to focusing on specific Raman bands, like the kinetics trend charts. Additionally, when compared to classification techniques like PLSDA or PCA, the results from the PLSR have a higher prediction accuracy, making it better suited for TSD estimations.

Overall, this study showed that Raman spectroscopy is a valuable tool for determining the time since deposition of bloodstains. There are other analytical methods that have been proposed for this purpose, including but not limited to IR spectroscopy, UV-vis spectroscopy, and liquid chromatography paired with mass spectrometry [15,38,39,40,41]. The scientific principles, advantages, and limitations of each technique are discussed in Table 5. In brief, several different techniques, which can offer information regarding the biochemical composition of blood, can be used to determine the time since deposition. While each technique offers unique advantages, Raman spectroscopy is one of the more versatile techniques for the forensic analysis of bloodstains. This method is more user friendly than liquid chromatography, requiring less expensive and specialized instrumentation. Additionally, water molecules are not Raman-active and therefore do not cause interference in the resulting spectrum, unlike with FTIR spectroscopy. Finally, Raman spectroscopy is a non-destructive technique, which means the samples can continue to be used for future analyses, such as DNA analysis. Overall, Raman spectroscopy is sensitive, selective, and non-destructive, making it an ideal technique for forensic analysis.

## 4. Conclusions

This proof-of-concept study showed that Raman spectroscopy is a versatile technique that is useful for the analysis of biological samples. Additionally, that it can be used to estimate the TSD of menstrual blood just as well as peripheral blood. While this research shows promise, testing this methodology with more donors is crucial before being able to be used by forensic practitioners. As we are already able to determine the TSD of peripheral bloodstains, it is necessary to show that this is possible for menstrual bloodstains as well. Menstrual blood’s composition is a combination of blood and vaginal components.

The goal of this preliminary research was to determine how the additional components of menstrual blood affect the aging process. To conduct this, menstrual blood was collected freshly from donors and aged up to two weeks post-deposition. For each trial, Raman spectra were collected at 1, 5, 9, 24, 48, 72, 96, 120, 144, 168, 264, and 336 h. The spectral results show changes that were apparent to the naked eye, indicating that there were biochemical alterations occurring during aging. The fluorescence background of the spectra exhibited an overall increase during the aging process, and the peak changes that occurred were consistent with previous aged peripheral blood results. Most notably, there was an increase in metHb peaks (374, 1369 cm^−1^), a decrease in oxyHb peaks (417 cm^−1^ and 1600 cm^−1^), and changes in the protein amide III band.

For the development of a quantitative mean for estimating the TSD, a chemometric analysis was performed. This involved the creation of a PCA model to estimate how the spectral data were clustered. While there was clustering based on the time point, there was a divide in the data that caused two large groupings of the spectral points. This indicated that any further classification analysis should be created using a binary model. Therefore, when the PLSDA and SMVDA binary models were created, it was conducted using a binary division into two classes: ≤72 h and ≥96 h. These models performed well and had an accuracy of 92% for the PLSDA. These models will allow investigators to rapidly have an approximation of whether the menstrual bloodstains are fresh (≤72 h) or whether they are older (≥96 h). Acting as a screening method, these classification models will allow analysts to understand if a further investigation is necessary for each stain.

Estimating a broad period of deposition (fresh vs. old) is an important starting point during an investigation. However, the capabilities of Raman spectroscopy when paired with chemometrics do allow for the calculation of a more specific TSD prediction. A PLSR model was created and had a R^2^_(CV)_ of 0.92 and R^2^_(Prediction)_ of 0.90. Using a regression model, forensic analysts can accurately predict the TSD of bloodstains. When combined, the classification and regression models can potentially serve as a form of presumptive and confirmatory tests for the TSD. This methodology allows for rapid, reliable, and non-destructive TSD estimations of bloodstains and proves that Raman spectroscopy is an ideal technique for forensic biological analysis. This is due to the large amount of information that can be ascertained when Raman spectroscopy is paired with chemometrics.

Even with the promising results to date, this is still a proof-of concept study. As such, the conclusions are currently limited by the number of samples tested to date and the time the blood was aged for. Furthering this research is a requirement before this method can be integrated into the current forensic workflow. Ideally, as more samples are tested, the aging models for both menstrual and peripheral blood will be able to be unified into a single universal model for bloodstain age determination. However, if this is not possible, then a hierarchical model can be created. This model would start with body fluid identification, followed by separate TSD determinations depending on the identification of the sample. Nonetheless, the creation of a larger donor pool and a longer post-deposition period would bolster the models’ capabilities. Additional future studies include, but are not limited to, testing the effects of the day of menstruation, the substrates the blood is deposited on, environmental conditions (temperature, humidity, and sunlight), and environmental contaminants on the aging process. These future studies will build on this proof-of-concept work to allow for the creation of a universal Raman spectroscopy method capable of determining the TSD of bloodstains.

## Figures and Tables

**Figure 1 sensors-24-03262-f001:**
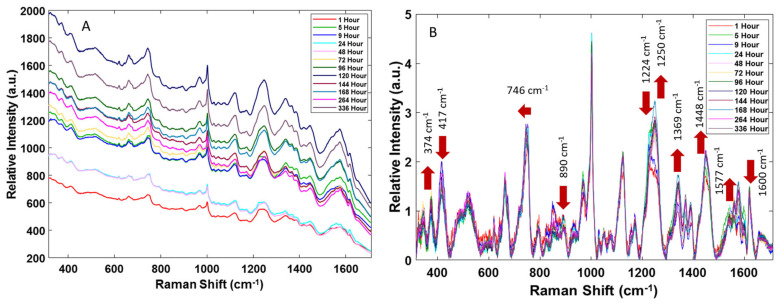
The averaged raw (**A**) and preprocessed (**B**) Raman spectra of menstrual blood. Spectra were collected between 1- and 336-h post-deposition. In (**B**), red arrows show the monotonic changes observed in the Raman spectra with TSD. Preprocessing includes the baseline corrected by the use of the AWF algorithm (lambda = 850, *p* = 0.001) and normalized by the total area.

**Figure 2 sensors-24-03262-f002:**
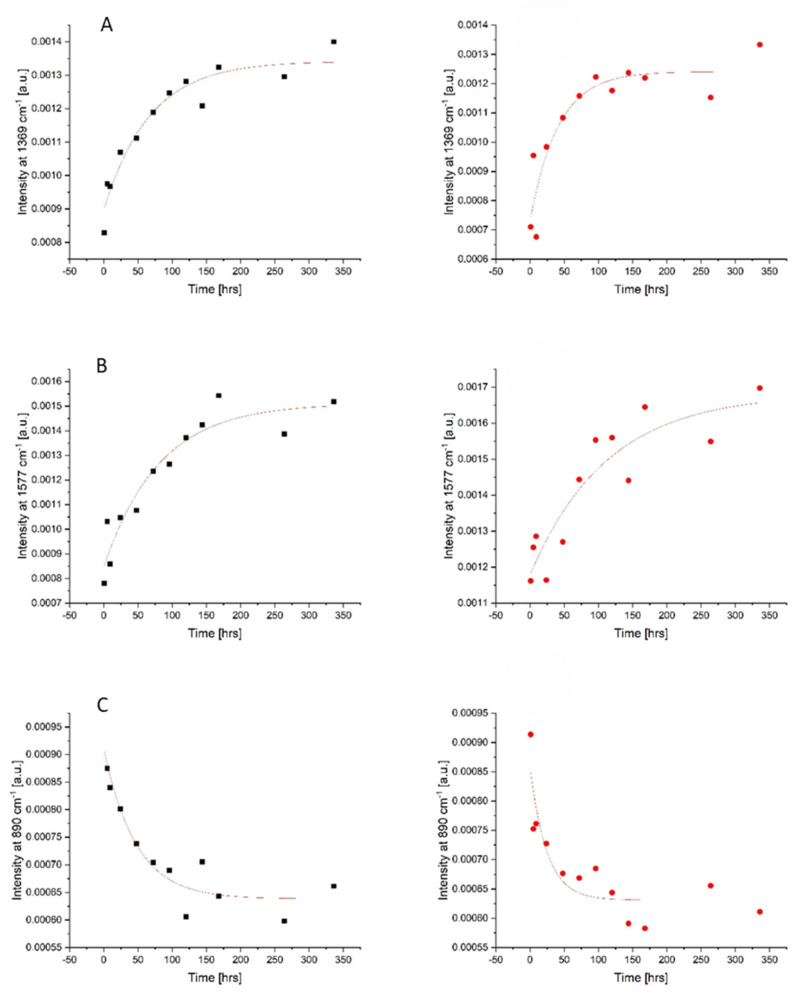
Intensity changes in the peaks at (**A**) 1369 cm^−1^, (**B**) 1577 cm^−1^, and (**C**) 890 cm^−1^ within 336 h after the deposition of the menstrual blood sample obtained from donors 1 (black) and 2 (red) along with the fitted exponential functions.

**Figure 3 sensors-24-03262-f003:**
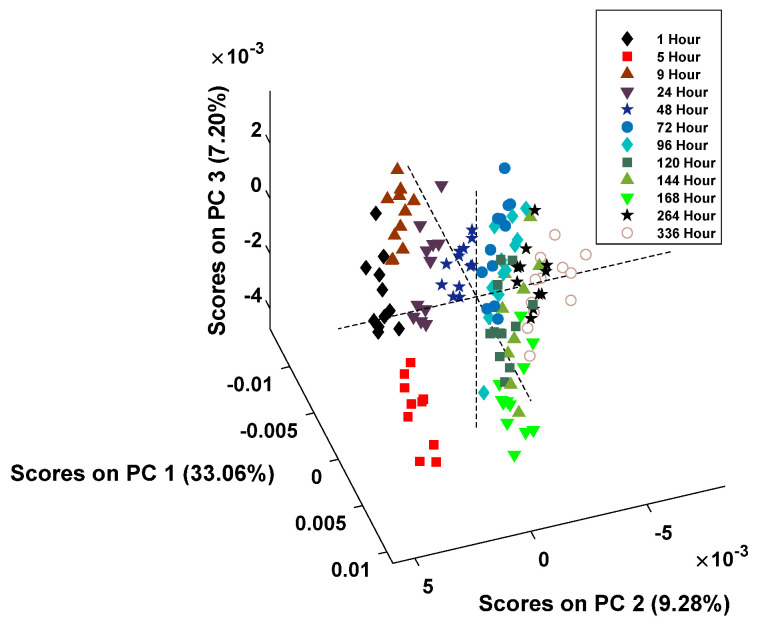
PCA score plot for donor 1 created with 4 principal components.

**Figure 4 sensors-24-03262-f004:**
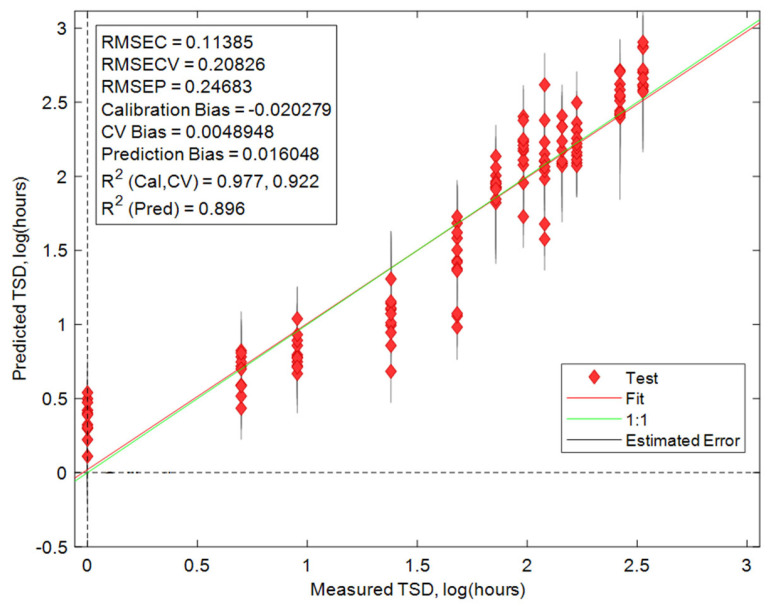
The PLSR model is presented in a logarithmic scale.

**Figure 5 sensors-24-03262-f005:**
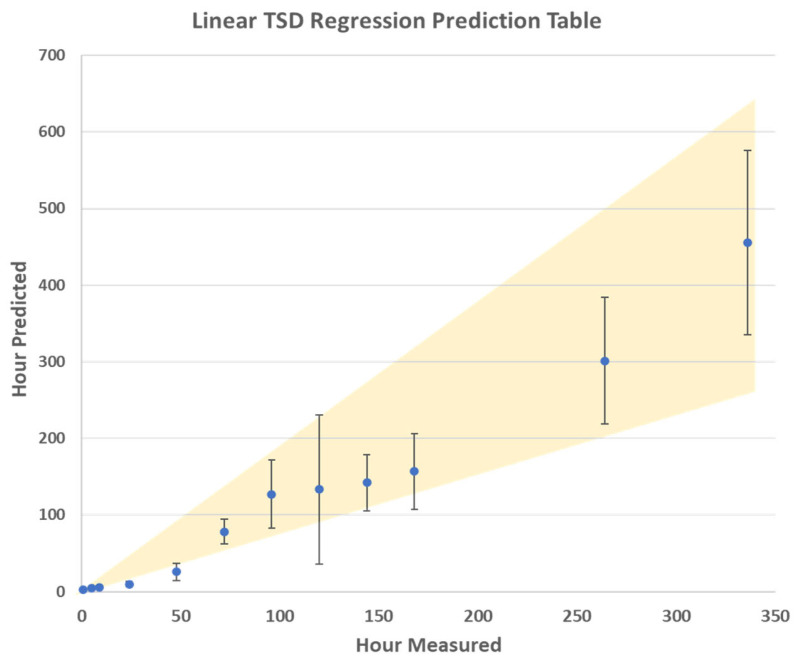
The average TSD predictions were also plotted on a linear scale with error bars showing one standard deviation.

**Table 1 sensors-24-03262-t001:** Spectral band assignments for menstrual blood. Components listed in red are associated with blood and components listed in blue are associated with vaginal fluid.

Raman Band/cm^−1^	Assignment	Refs.
374	Heme (metHb)	[22]
417	Heme (oxyHB)	[22]
518, 663, 794	Heme	[23,32]
746	Tryptophan	[3]
852–890	Amino acids, lactate	[3]
935	Acetic acid	[3]
970	Proteins, lipids	[3]
1003	Phenylalanine	[3]
1050	Lactic acid, urea, proteins	[3]
1075	Lactic acid	[3]
1124	Heme, lactic acid, urea	[3]
1171	Heme	[32]
1248	Proteins (amide III), guanine, cytosine	[3]
1341	Tryptophan	[3]
1369	Heme	[3]
1389	Urea	[3]
1448	Tryptophan, lactic acid, urea	[3]
1540	Heme	[3]
1577	Proteins, DNA bases	[3]
1619	Heme	[3]
1660	Proteins (amide I), lactic acid, urea	[3]

**Table 2 sensors-24-03262-t002:** The parameters of the fitted exponential functions (t_1_, R^2^) for the intensity changes at 1369 cm^−1^, (B) 1577 cm^−1^, and (C) 890 cm^−1^.

Peak Position (cm^−1^)	Characteristic Time (t_1_ ± Error) (h)		R^2^	
	D1	D2	D1	D2
890	47p ± 10	25 ± 10	0.92	0.81
1369	70 ± 20	40 ± 20	0.93	0.85
1577	80 ± 30	110 ± 50	0.91	0.84

**Table 3 sensors-24-03262-t003:** Binary Partial Least Squares Discriminate Analysis Model Prediction Results for Bloodstains.

**Model Results**	**≤72 h**	**>72 h**
Predicted as ≤ 72 h	65	1
Predicted as >72 h	5	67
Predicted as Unassigned	0	0
Percent Prediction	93%	99%
**Internal CV Results**	**≤72 h**	**>72 h**
Predicted as ≤ 72 h	60	1
Predicted as >72 h	11	67
Predicted as Unassigned	0	0
Percent Prediction	85%	99%
**External Validation Results**	**≤72 h**	**>72 h**
Predicted as ≤72 h	60	5
Predicted as >72 h	7	63
Predicted as Unassigned	0	0
Percent Prediction	90%	93%

**Table 4 sensors-24-03262-t004:** Classification of the TSD by spectra. The sample at the 72-h time point was misclassified (text in red). This indicates that over 40% of the spectra were misclassified.

External Validation Results by Hour	Predicted as ≤72 h	Predicted as >72 h
1 h	11	0
5 h	10	0
9 h	11	0
24 h	11	0
48 h	12	0
72 h	5	7
96 h	4	8
120 h	1	11
144 h	0	9
168 h	0	12
264 h	0	11
336 h	0	12

**Table 5 sensors-24-03262-t005:** Comparison of some spectroscopic and chromatographic techniques for determining the time since deposition of bloodstains.

Technique	Scientific Principle	Advantages	Limitations	References
Raman Spectroscopy	Analyzes inelastically scattered light providing information on molecular vibrations.	High specificity and sensitivity, potentially allowing for the detection of molecular changes in bloodstains over time.	Might not be applicable for a crime scene scenario where bloodstains have been cleaned.	[15,38]
Infrared Spectroscopy	Analyzes the absorption of infrared radiation by molecules in bloodstains, providing information on molecular vibrations.	Can detect changes in chemical composition and structure over time.	Water in the sample can interfere with the FTIR spectra of essential blood components.	[15,38]
UV–Visible Spectroscopy	Measures the absorption of light in the UV–visible range by compounds present in bloodstains.	Can provide information on the degradation of specific compounds over time.	The results are highly dependent on the environmental conditions the bloodstain was deposited in (e.g., humidity and temperature).	[39]
Liquid Chromatography	Separates the components of a bloodstain sample using a liquid mobile phase and a solid or liquid stationary phase.	Can be employed to separate and analyze non-volatile compounds in bloodstains to identify changes in metabolic markers in blood.	Requires specialized equipment and expertise (e.g., triple quadrupole MS or quantitative time-of-flight MS).	[40,41]

## Data Availability

Data are contained within the article.

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
