# Peer review of "Raman Spectroscopy for the Time since Deposition Estimation of a Menstrual Bloodstain"

_sensors, 2024, doi:10.3390/s24113262_

Round 1

Reviewer 1 Report

Comments and Suggestions for Authors

The reviewed paper explores the pivotal role of Raman spectroscopy in forensic chemistry, specifically in the analysis of bloodstains, where distinguishing between peripheral and menstrual bloodstains poses a significant challenge. By demonstrating Raman spectroscopy's effectiveness in discriminating between these types of bloodstains and providing models for estimating the time since deposition (TSD) of menstrual blood, the study fills a crucial gap in forensic analysis. The findings not only offer invaluable probative information but also pave the way for potential implementation in forensic practice, with the aim of developing universal models for bloodstain age determination, thus enhancing the accuracy of forensic investigations.

The paper provides a concise and comprehensive explanation of the utilized Raman spectroscopy process and the analysis of blood sample peaks. The thorough analysis of spectral data enhances understanding and underscores the significance of Raman spectroscopy in forensic science, showcasing its potential for precise and reliable bloodstain analysis.

1.        There are a few issues with the mapping data collection method outlined in the paper. While the method mentions collecting data at "12 map points", it fails to explain how these points were selected, the area they covered, or whether they adequately represent the sample's heterogeneity. Additionally, the method describes analyzing individual spectra and removing low-quality spectra based on significant deviation from an average spectrum, but it does not specify the criteria for determining what constitutes a "low-quality" spectrum, especially in the context of heterogeneity analysis, or how much deviation is considered significant. Furthermore, the method implies that spectra were removed based on experimental imperfections like poor focus on the bloodstain, which introduces subjective decision-making and potential bias into the data analysis process. Addressing these issues by providing more details on the selection process of map points and the criteria for identifying and eliminating low-quality spectra would enhance the reliability and transparency of the data collection procedure.

2.        The observed increase in intensity of certain peaks, such as those at 1124 cm-1 and 1448 cm-1, in Figure 1A-B over time raises questions about the underlying mechanisms. Given the testing period exceeding 1 hour, it seems unlikely that the increase is solely attributable to concentration changes resulting from water loss. Further investigation into the molecular dynamics underlying these spectral changes could provide valuable insights into the observed increase in peak intensity, despite the stability of other spectral features.

3.        I believe this technique, especially if utilizing portable Raman spectroscopes, could serve as a valuable presumptive test. While Raman spectroscopy offers valuable insights into the composition and characteristics of bloodstains, it may not entirely supplant DNA analysis in every scenario. DNA analysis remains a highly reliable method for individual identification and can furnish critical evidence in forensic investigations, particularly when establishing definitive individual identities is essential. Therefore, advocating for this method as a cost-effective substitute for confirmatory tests like DNA analysis may not be appropriate.

4.        I highly encourage authors to include a new table at the end of the discussion section, comparing their developed technique with other possible similar methods (such as spectrofluorometry, Raman spectroscopy, chromatography-based approaches, and electrochemical methods) for TSD and blood identification. Such a table would offer readers a clear and concise overview of the strengths and limitations of this technique, facilitating a better understanding of the comparative performance and applicability in forensic contexts. In the discussion, a brief summary of this table and comparison would provide additional context for interpreting the findings and underscore the significance of the developed technique in advancing the field of bloodstain analysis.

Reviewer 2 Report

Comments and Suggestions for Authors

The work takes up a very important issue in forensic science and points to the key role of experimentation when analysing/interpreting the results of research during forensic examinations.

However, I would like to point out that despite the high practical value of the work and the invaluable influence of the analysed research results on the correct interpretation of the evidence, several changes should be made to the work in order to increase its scientific value.

I recommend:

1.      Rewriting the abstract in terms of relevant information related to the applied research method and research material, i.e. specifying already in the abstract, which blood component, substance is proposed in the differential analysis between peripheral and menstrual blood, as relevant during the analysis of the Raman scattering spectrum and further applied statistical analysis model.

2.      The proposal is to remove the sentence or make it more precise, as the methods currently used are counted, and indeed the literature should be cited.

„ Among the myriad of analytical methods uti-11 lized, the use of Raman spectroscopy for body fluid traces, notably bloodstains, has proven to be 12 versatile.”

3.      The first sentence in the abstract and introduction is the same, proposal to remove the repetition.

4.      „This disci-29 pline is the application of chemistry to assist in solving forensic investigations conducted 30 by law enforcement. In this field, several analytical methods are used, including those for 31 the analysis of body fluid traces. „

Lack of literature, it is not necessary to name the methods, but a reference to the literature is.

5.      Line 39 to 46, does only literature [2] describe, please increase the number of citations in this field.

6.      „Additional information that can help investigators reconstruct the event in question, 46 is by placing the evidence temporally at the scene.”

 this sentence is not precise, the temporal location of the evidence at the scene is not useful, but to carry out an experiment, with the material being the reference of the evidence, the pattern of the evidence is the correct notation.

7.      A wonderful scientific manuscript piece on the composition and difference between peripheral and menstrual blood. The paragraph clearly tips the complex interpretation of the Raman scattering spectrum studied. The authors of this paper should clearly pay attention to the chemical composition of the analysed samples, and analyse the vibrations between atoms visible on the Raman scattering spectrum, and this will lead to the postulation of dominant structures on the presented spectrum. Unfortunately, due to my extensive practical experience, I cannot agree with a huge fragment in the content, which leads the authors to differentiate analysed samples due to the influence of fluorescence. This interpretation is not scientifically correct, because the authors chose a laser above the UV/VIS range (785 nm) and excitation of fluorescently active blood components was eliminated, while the lack of homogeneity and repeatability between the analysed samples in terms of composition results in a background coming from a multicomponent matrix, affecting the quality of Raman scattering imaging. Therefore, I propose to work in the future on the deconvolution of the spectrum and the subtraction of the background, and not to mix fluorescence in the interpretation of the Raman scattering spectrum in such a categorical way. A suggestion to improve/replace the section on fluorescence to the influence of sample multicomponent and inhomogeneity, which the authors of the paper describe beautifully in the rest of the paper.

8.      „As blood is a complex matrix it can be difficult to identify a singular chemical phe-301 nomenon responsible for changes in an individual Raman peak with TSD.”

This sentence is very truthful and should be the guiding force in the discussion, and an attempt made to take the characteristics of a particular type of vibration from a particular stable compound.......

The manuscript is recommended for publication after taking into account the above suggestions.

Round 2

Reviewer 1 Report

Comments and Suggestions for Authors

Dear Authors,

Thank you for your efforts in revising the manuscript. While I appreciate the improvements made, I must express some concerns. Out of the four comments I provided, only two (1 and 3) were adequately addressed. Comments 2 and 4 regarding the FTIR spectra and the comparison table were not addressed effectively.

Regarding comment 2, your response does not fully clarify the changes observed in the FTIR spectra. For example, the increase in intensity of polysaccharides' peaks doesn't correlate with concentration changes, and the alterations in the fluorescence spectra of tryptophan shouldn't be directly related to the concentration increase of its FTIR peak. If there is an interaction with another molecule (tryptophan derivative) leading to a peak at 1448 cm-1, this should be clearly explained along with a discussion of the molecule produced. 

As for comment 4, my apologies if I was unclear. I was expecting a comparison of your detection technique with other published methods in the field. Even with the current Table 5, it would needed to include references supporting your arguments regarding the limitations and advantages of each technique.

Overall, I encourage you to further refine these aspects to strengthen the scientific rigor and clarity of your work.

Author Response

Please find uploaded our response letter.
